# Correlation of Systemic Inflammation Parameters and Serum SLFN11 in Small Cell Lung Cancer—A Prospective Pilot Study

**DOI:** 10.3390/biomedicines12050976

**Published:** 2024-04-29

**Authors:** Ivana Simić, Azra Guzonjić, Jelena Kotur Stevuljević, Vesna Ćeriman Krstić, Natalija Samardžić, Katarina Savić Vujović, Dragana Jovanović

**Affiliations:** 1Medical Affairs, Merck Sharp & Dohme d.o.o., Omladinskih brigada 90a, 11070 Belgrade, Serbia; 2Department for Medical Biochemistry, Faculty of Pharmacy, University of Belgrade, Vojvode Stepe 450, 11221 Belgrade, Serbia; azra.guzonjic@pharmacy.bg.ac.rs (A.G.); jelena.kotur@pharmacy.bg.ac.rs (J.K.S.); 3Faculty of Medicine, University of Belgrade, Dr Subotica 8, 11000 Belgrade, Serbia; vesna.ceriman@kcs.ac.rs; 4Clinic for Pulmonology, University Clinical Center of Serbia, Dr Koste Todorovica 26, 11000 Belgrade, Serbia; natalija.samardzic@kcs.ac.rs; 5Department of Pharmacology, Clinical Pharmacology and Toxicology, Faculty of Medicine, University of Belgrade, Dr Subotica 8, 11000 Belgrade, Serbia; katarinasavicvujovic@gmail.com; 6Internal Medicine Clinic “Akta Medica”, Cara Nikolaja II, 11000 Belgrade, Serbia; draganajv@yahoo.com

**Keywords:** biomarker, inflammation, sPD-L1, SLFN11, SCLC

## Abstract

Background and objectives: The objective of this research was to analyze the correlation of the neutrophil-to-lymphocyte ratio (NLR), C-reactive protein (CRP), soluble programmed cell death ligand 1 (sPD-L1), and Schlafen 11 (SLFN11) with the response to first-line chemotherapy in a cohort of small cell lung cancer (SCLC) patients, and to determine their potential as predictive serum biomarkers. Materials and Methods: A total of 60 SCLC patients were included. Blood samples were taken to determine CRP, sPD-L1, and SLFN11 levels. The first sampling was performed before the start of chemotherapy, the second after two cycles, and the third after four cycles of chemotherapy. Results: The patients who died earlier during the study had NLR and SLFN11 concentrations significantly higher compared to the survivor group. In the group of survivors, after two cycles of chemotherapy, the NLR ratio decreased significantly (*p* < 0.01), but after four cycles, the NLR ratio increased (*p* < 0.05). Their serum SLFN11 concentration increased significantly (*p* < 0.001) after two cycles of chemotherapy, but after four cycles, the level of SLFN11 fell significantly (*p* < 0.01). CRP, NLR, and SLFN11 were significant predictors of patient survival according to Kaplan–Meier analysis. The combination of inflammatory parameters and SLFN11 with a cutoff value above the 75th percentile of the predicted probability was associated with significantly lower overall survival in SCLC patients (average survival of 3.6 months vs. 4.8 months). Conclusion: The combination of inflammatory markers and the levels of two specific proteins (sPD-L1, SLFN11) could potentially serve as a non-invasive biomarker for predicting responses to DNA-damaging therapeutic agents in SCLC.

## 1. Introduction

Lung cancer accounts for approximately 12% to 13% of newly identified cancer cases globally, which is more than 180,000 cases per year. In terms of frequency of occurrence, it is the second most common cancer, and it is also the leading cause of death from malignant diseases [1]. Small cell lung cancer (SCLC) is characterized by a poorly understood underlying pathophysiology and an exceedingly bleak outlook. Managing it necessitates a multifaceted approach encompassing clinical evaluation, diagnosis, and treatment. It is worth noting that a significant portion of individuals diagnosed with SCLC have a smoking history, whether current or former [2]. Although less frequent in non-smokers, unlike non-small cell lung cancer (NSCLC), SCLC does not seem to be associated with specific somatic mutations. SCLC is characterized by its rapid progression, involving swift growth and early dissemination, including spread to distant organs. Approximately 30% of patients are identified with LS SCLC, according to the VALSG system [2]. For the last 30 years, the backbone of treatment for SCLC has been etoposide and platinum-based chemotherapy. Most patients experience a relapse within 6 to 9 months following the completion of their initial treatment, resulting in a median survival of 16 to 24 months for those with limited disease (LD) and 6 to 12 months for those with extensive disease (ED) [2]. Immunotherapy was incorporated into first-line treatment for SCLC after 2018 [3].

Recent studies profiling SCLC have revealed different molecular subtypes based on the expression of specific transcriptional regulators, offering insights into potential subtype-specific therapeutic approaches. These subtypes are defined by the relative expression of four key transcriptional regulators: ASCL1-high (SCLC-A), NEUROD1-high (SCLC-N), POU2F3-high (SCLC-P), and YAP1-high (SCLC-Y) [2]. These transcription factors (TFs) are associated with distinct expression profiles of neuroendocrine (NE) markers and could direct research into tailored therapeutic targets for each subtype. The SCLC-A (ASCL1) subtype exhibits a high expression of ASCL1 and NE markers and has a classic morphology. In contrast, the NE-high subtype, called SCLC-N (NEUROD1), exhibits a distinct morphology [2].

SCLCs that have low or no expression of NEUROD1, ASCL1, and INSM1 are categorized as either SCLC-Y (YAP1) or SCLC-P (POU2F3) subtype based on their TF expression pattern [4]. YAP1 is a transcription regulator activated by the HIPPO signaling pathway, while POU2F3 is essential for the development of pulmonary tuft cells and chemosensory cells in the gastrointestinal epithelium [5,6]. Recent data suggest that SCLC-P subtype tumors may originate from tuft cells due to their high POU2F3 expression patterns [4,6]. These developments are promising and underscore the importance of continuing the search for biomarkers that can assist in selecting the most appropriate therapy tailored to the specific subtype of SCLC.

Unlike patients with advanced or recurrent NSCLC, whose personalized therapy is guided by a range of oncogenic driver mutations and PD-L1 levels, identifying biomarkers and therapeutic targets in SCLC continues to pose a challenge. The SCLC genome exhibits an exceptionally high mutation rate [2]. The most common SCLC mutations, i.e., tumor suppressor genes *TP53* and *RB1*, are currently not druggable targets [7]. Frequent alterations include *MYC* family gene changes, common amplifications, and inactivating mutations in *NOTCH* family genes. Other genetic alterations identified include *PTEN* loss and *FGFR1* amplifications [7].

A significant recent discovery in preclinical studies involves SLFN11, a protein commonly referred to as Schlafen 11. SLFN11 is recognized as a mediator in the DNA damage response and can lead to an irreversible block of DNA replication. SLFN11 has the potential to be a crucial biomarker in the choice of chemotherapy agents, including commonly used drugs like platinum-based compounds, for the treatment of SCLC [8]. SLFN11 belongs to the family of genes that play a role in cell cycle regulation, immune response, and tumor suppression. The precise mechanism through which SLFN11 impacts sensitivity to chemotherapy is not completely clear. Nevertheless, it is hypothesized that SLFN11 may influence DNA replication and repair mechanisms, making cancer cells more susceptible to the effects of DNA-damaging chemotherapy agents. When SCLC tumors show elevated levels of SLFN11 expression, they often demonstrate increased sensitivity to chemotherapy drugs, resulting in improved treatment responses. Conversely, SCLC tumors with low or no SLFN11 expression are more prone to be resistant to chemotherapy and are associated with a less favorable prognosis [9].

Another intriguing subject involves numerous publications suggesting that factors related to systemic inflammation, oxidative status, and dyslipidemia may have a role in both the development and clinical progression of cancer. One marker of systemic inflammation that has garnered attention in the context of SCLC is the neutrophil-to-lymphocyte ratio (NLR). This ratio is calculated by dividing the absolute neutrophil count by the absolute lymphocyte count in a patient’s blood sample. Elevated NLR levels have been linked to unfavorable outcomes in SCLC. Another marker of systemic inflammation that has been explored in SCLC is CRP. Elevated CRP levels have also been associated with adverse outcomes in SCLC patients [10].

The PD-1/PD-L1 pathway is widely acknowledged for its role in regulating immune tolerance within the tumor microenvironment. PD-1 and its ligands, PD-L1 and PD-L2, play a crucial role in controlling T-cell activation, proliferation, and cytotoxic activity, ultimately affecting the anti-tumor immune response [11]. While tumor PD-L1 expression serves as a predictive marker for clinical responses to anti-PD-1/PD-L1 therapy in NSCLC, its role and association with clinicopathological characteristics, including treatment outcomes and prognosis, in high-grade neuroendocrine tumors such as SCLC remain relatively ambiguous and underexplored [2].

The objectives of this research are as follows:(a)The original aspect of this novel research lies in determining the serum level of SLFN11 in a cohort of SCLC patients undergoing chemotherapy and exploring its potential correlation with treatment response and survival. This marks the first study of its kind in this context;(b)To define the serum levels of systemic inflammation parameters NLR and CRP and of soluble PD-L1;(c)To analyze the correlation between NLR, serum concentrations of CRP, soluble PD-L1, and SLFN11 with the outcome and the response to first-line chemotherapy in the cohort of SCLC patients, and to explore the potential of each of these parameters, as well as their different combinations, and whether they can serve as predictive serum biomarkers in SCLC patients.

## 2. Materials and Methods

### 2.1. Research Ethics Considerations

All patients were enrolled in this study between October 2020 and February 2022 at a single center and were treated with platinum-based chemotherapy. The study was approved by the School of Medicine, Belgrade University Ethics and Review Board (No. 1322/II-81); all patients provided written informed consent.

### 2.2. Patient Selection Criteria

This study included patients with pathophysiologically confirmed SCLC in the locally advanced stage III (IIIa, IIIb, IIIc) categorized as limited-stage (LS) SCLC, as well as those at the metastatic stage IV associated with extensive-stage (ES) SCLC. Exclusion criteria were SCLC patients younger than 18 years and patients with ECOG PS (Eastern Cooperative Oncology Group Performance Status) 2–4. In total, 60 consecutive patients with SCLC were included, all of whom were treated with the first-line standard chemotherapy, a combination of platinum and etoposide, at the University Hospital of Pulmonology, Clinical Center of Serbia, Belgrade. Due to technical limitations at the University Hospital of Pulmonology, Clinical Center of Serbia, concurrent radiotherapy was not available. As a result, all patients with stage III disease underwent sequential chemoradiotherapy, with radiotherapy being administered after the completion of the 4th cycle. Patient and tumor baseline data were recorded at the initial examination.

### 2.3. Sample Collection and Processing

Blood samples from this group of patients were collected in the morning hours from the anterior cubital vein in 6 mL vacutainer systems (one with a serum separator gel for the serum sample and the second with an EDTA anticoagulant for the plasma sample) (BD Diagnostics, Wokingham, UK). The first sampling was performed before the initiation of chemotherapy, the second after two cycles of therapy, and the third blood sampling after the administration of four cycles of chemotherapy. The blood samples were collected at multiple time points to monitor the dynamic changes in inflammatory parameters (CRP and NLR), SLFN11 levels, and soluble PD-L1 levels. These measurements, taken at baseline and after the second and fourth cycles of chemotherapy, offered insights into the impact of treatment on biomarker levels. They also shed light on potential associations with treatment response and survival, enabling us to evaluate their predictive value over time.

Routine hematology parameters analyzed in this study (leukocyte count, neutrophil and lymphocyte percentages) were determined according to standard laboratory protocols. The NLR was subsequently calculated as a ratio of the neutrophil and lymphocyte percentages. CRP concentration was measured by an immunoturbidimetric method. For the sPD-L1 determination in human plasma, the DuoSet ELISA system (R&D Systems Europe, Ltd., Abingdon, UK) was used as a sandwich enzyme-linked immunosorbent assay (ELISA). The SLFN11 concentration in serum was determined using sandwich ELISA technology (Wuhan Fine Biotech Co., Ltd., Wuhan, Hubei, China). Considering that most cancer patients typically exhibit elevated inflammation parameters (such as CRP and NLR ratio), we opted not to use normal values as a reference. Instead, based on our statistical analysis, we set the 75th percentile as the high-risk cutoff value for CRP and NLR. It is worth noting that there is no consensus regarding the reference range for SLFN11 and PD-L1. Therefore, we also chose the 75th percentile as the high-risk cutoff value for these parameters.

### 2.4. Statistical Methods

The Kolmogorov–Smirnov test was used to check the normality of the distribution of parameters. According to this, test data are presented as means ± standard deviations or median values and interquartile ranges (25th–75th percentile) for continuous data. Categorical data are presented as numbers (percentages). A comparison of related values was performed using Friedman’s test followed by Wilcoxon’s paired test as a post hoc test. We used a Kruskal–Wallis non-parametric ANOVA with the Mann–Whitney U test as a post hoc test for inter-group comparison. Kaplan–Meier survival analysis was used to estimate the survival function concerning the risk values of selected parameters. Binary logistic regression analysis was utilized to integrate two or more parameters into models through predicted probability calculation.

## 3. Results

### 3.1. Baseline Characteristics

Baseline characteristics of 60 SCLC patients—age, gender, smoking status, disease stage, TNM staging, distant metastasis, and survival—are presented in Table 1.

All patients included in this study were of Caucasian ethnicity. Most patients were older than 60, with a predominance of male patients (61.6%), and almost all were current or ex-smokers. More than half of the patients were categorized as having limited-stage (LS) disease, and all of them had an ECOG PS of 0–1. Platinum–etoposide doublet chemotherapy was the first-line therapy in all patients, whereas several patients were treated with concurrent IO or with palliative radiotherapy as well. After two cycles of chemotherapy, 83.3% of patients were alive. However, at the end of the study follow-up period, 41.7% of the patients were alive after four cycles.

### 3.2. The Dynamics of Inflammatory Parameter Values and sPD-L1 and SLFN11 Concentration Changes in Different Subgroups during the Study

Inflammatory marker (NLR and CRP) values and serum concentrations of two specific proteins—soluble PD-L1 and SLFN11—were determined at three study points: baseline, after two, and after four chemotherapy cycles (Table 2). In the survivor group, there was a statistically significant difference in the level of sPD-L1 (*p* = 0.011), with a higher level observed in females. Also, we found statistically significantly higher levels of SLFN11 in females from the deceased group, among those who died within the first 2 months (*p* = 0.030). Additionally, in the deceased group, the levels of SLFN11 remained statistically significantly higher after two cycles of chemotherapy (*p* = 0.017).

Most of the patients died during the study period at different time points. Accordingly, we have categorized several subgroups based on their survival time and the timing of their death (Table 2).

The results show that patients who died earlier during this study, i.e., the course of the disease, initially had the highest value of CRP and the highest neutrophil percentage but the lowest percentage of lymphocytes. Therefore, their NLR was significantly higher compared to the survivor group. This group of patients also had a higher SLFN11 concentration compared to the survivors. When we compared the baseline levels of inflammatory parameters, SLFN11 and sPD-L1, in survivors vs. the deceased in the 3rd-4th month of the study period, no statistical significance was observed between these groups (Table 2).

In the survivor group, the leukocyte count and neutrophil percentage showed a significant decrease after two cycles of chemotherapy, whereas the lymphocyte percentage significantly increased. Consequently, their NLR decreased significantly. We observed a significant increase in the percentage of neutrophils and NLR after four cycles of chemotherapy, compared to the values after two cycles of chemotherapy in this group. After four cycles of chemotherapy, CRP levels were also significantly decreased.

A significant increase in the serum concentration of SLFN11 after two cycles of chemotherapy was observed in the survivor group, but after four cycles, the level of SLFN11 fell significantly; however, it was significantly higher than the baseline. We also found a statistically significant increase in sPD-L1 levels after two cycles of chemotherapy. Although the level of sPD-L1 remained higher after four cycles of chemotherapy, this value was not statistically significant (Table 2). In the group of deceased patients, a significant increase in SLFN11 serum levels was evidenced after two cycles of chemotherapy (Table 2).

### 3.3. Survival Analysis Based on Selected Inflammatory Markers

To test the predictive capability of selected parameters for overall patient survival, we used the Kaplan–Meier survival analysis for 6 months of the study period for CRP, NLR, and SLFN11 according to distinct cutoff values determined for any of the three parameters as values above the 75th percentile values for our current study patient group. CRP, NLR, and SLFN11 were significant predictors of patient survival according to the Kaplan–Meier analysis (Figure 1, Figure 2 and Figure 3).

To obtain better mortality prediction, we used binary logistic regression to model individual parameters from the preliminary analysis. Kaplan–Meier analysis was used for combined parameters, i.e., the CRP and NLR combination (inflammatory model) (Figure 4), as well as the CRP, NLR, and SLFN11 combination (inflammatory biomarkers plus the SLFN11 model) (Figure 5).

The results presented in Figure 1 show that the survival of patients with CRP levels above 50 mg/L was significantly shorter compared to patients with lower CRP values. Figure 2 shows that an NLR above 4.9 was an indicator for shorter patient survival. The results in Figure 3 indicate that a level of SLFN11 higher than 607 ng/mL predicted poorer overall survival in this group of patients.

In this analysis, combining inflammatory parameters with the 75th percentile of predicted probability as the cutoff value revealed a significantly lower overall survival among SCLC patients. On average, patients with values above the 75th percentile had a survival of 3.1 months, compared to 4.9 months for those with values below the 75th percentile. Also, inflammatory parameters and the SLFN11 combination as potential predictive biomarkers showed that the patients with higher levels of this biomarker had 3.6 months of overall survival compared to 4.8 months in patients with lower levels of this integrated biomarker (Table 3).

Further analysis regarding NLR risk value subgroups (75th percentile of NLR as cutoff value) revealed significant differences only in CRP, WBCs, and SLFN11. Patients with high-risk NLR values had, at the same time, significantly increased CRP, leukocyte count, and SLFN11 concentration compared to patients with an NLR below the 75th percentile (Table 4).

## 4. Discussion

The results of this study revealed significant changes in different biomarkers during the research period. Throughout the study, there was a notable decrease in CRP concentration and leukocyte and neutrophil counts, along with an increase in lymphocyte count from baseline after two and four cycles of chemotherapy. This led to a significant decrease in the NLR from baseline to the point after two cycles of chemotherapy. SLFN11 levels significantly increased from baseline to the point after two cycles of chemotherapy but decreased significantly after four cycles of therapy. Another protein, sPD-L1, significantly increased from baseline to the point after two cycles of chemotherapy and continued to rise after four chemotherapy cycles, although this increase did not reach statistical significance. Patients with shorter overall survival initially exhibited significantly higher NLR values compared to survivors. This group of patients also had higher SLFN11 concentrations. According to the Kaplan–Meier analysis, our study showed the predictive capability of certain parameters for overall survival, such as CRP, NLR, and SLFN11. Also, combining inflammatory parameters (CRP and NLR) showed significantly lower overall survival for SCLC patients, as did the combination of inflammatory parameters and SLFN11.

At present, the two most reliable prognostic factors related to SCLC during diagnosis are the disease stage and ECOG PS. However, it is important to note that even within the same disease stage and ECOG PS category, substantial variations in prognosis can exist. Therefore, there is a critical need for additional, preferably serum-based, prognostic markers to improve the accuracy of treatment response prediction and prognosis of SCLC patients.

Recent profiling of SCLC findings, offering insights into the molecular subtypes of SCLC, represents a significant advancement in our understanding of SCLC biology [7].

This newfound understanding opens doors to exploring alternative treatment approaches tailored to the specific characteristics of each subtype. By targeting the unique features and vulnerabilities of each SCLC subtype, researchers and clinicians could develop more effective and personalized treatment strategies. New potential biomarkers should be explored further.

Published data on the SLFN11 expression level in SCLC are limited, but it appears that assessing the SLFN11 expression level in SCLC tumors could help in determining the appropriate treatment strategy. Overall, SLFN biomarkers, particularly SLFN11, possess promising clinical utility in SCLC by offering valuable information for personalized treatment decisions. This is particularly significant given that many SCLC patients may lack sufficient archival tissue material for biomarker testing. Therefore, staining for SLFN11 by immunohistochemistry (ICH) may not always be feasible [12]. On the other hand, liquid biopsy or circulating tumor cells could complement tissue biopsy. Zhang and colleagues [13] showed that SLFN11 can be detected in circulating SCLC tumor cells collected from a blood draw. SLFN11 expression levels change dynamically during treatment: while 70% of circulating tumor cells expressed SLFN11 in treatment-naive patients, this proportion decreased to 25% in patients receiving platinum therapy. This observation suggests that the level of SLFN11 may change over the course of treatment, but also suggests the possibility to assess the status of the biomarker at the time of disease progression. One of the potential mechanisms identified for SLFN11 downregulation is hypermethylation of its promoter region [14]. In preclinical models, epigenetic modifiers that reverse SLFN11 promoter methylation, such as EZH2 inhibitors, have initiated the re-expression of SLFN11, which led to sensitization to DNA-damaging agents [15]. Based on these preclinical findings, a phase I/II trial investigating the combination of EZH1/2 inhibitor DS-3201b and irinotecan in recurrent SCLC is ongoing (NCT03879798).

SLFN11 expression has been linked to the sensitivity of cells to various DNA-damaging drugs, including topotecan, cisplatin, and irinotecan. SLFN11 is involved in the regulation of biological functions such as cell proliferation, immune responses, and viral replication [14]. In the context of cancer, SLFN11 helps sensitize cancer cells to DNA-damaging agents like topoisomerase I and II inhibitors (such as irinotecan and etoposide, respectively), DNA synthesis inhibitors (e.g., gemcitabine), and DNA cross-linkers and alkylating agents (e.g., cisplatin) [8,14]. The specific function of the DNA/RNA helicase-like motif in the C-terminal region of SLFN11 is still unknown, but similar motifs have been associated with crucial roles in the cellular response to DNA damage [14]. Previous studies have suggested that SLFN11 interacts with key proteins involved in the DNA damage response system, such as replication proteins RPA1, RPA2, and RPA3, as well as BRCA1-associated ring domain protein (BARD1). Hence, it can be speculated that epigenetic silencing of SLFN11 disrupts the proper interaction between DHX9 and BRCA1, leading to functional changes in the DNA damage response system. This alteration may ultimately affect the sensitivity of cancer cells to platinum-based chemotherapy.

Winkler et al. showed that, contrary to the previous findings, SLFN11 protein levels did not decrease after chemotherapy treatment [16]. However, previous studies have demonstrated a correlation between SLFN11 hypermethylation and decreased overall survival (OS) in a cohort of NSCLC patients [14]. In our study, we observed that patients with a shorter survival time had higher levels of SLFN11. Interestingly, the concentration of SLFN11 significantly increased after two cycles of chemotherapy. However, in the survivor group, it decreased after four cycles of chemotherapy. Nonetheless, even after four cycles of chemotherapy, the level of SLFN11 remained higher than the initial baseline value. Therefore, the increase in SLFN11 levels observed during platinum-based therapy may be due to the activation of the DNA damage response pathways, replication stress in cancer cells, and the cells’ attempt to repair the chemotherapy-induced DNA damage. This upregulation of SLFN11 may contribute to the sensitivity of cancer cells to platinum-based drugs and their subsequent therapeutic effects [17]. Furthermore, we observed a decrease in SLFN11 expression after four cycles of chemotherapy. Several studies confirmed that the loss of SLFN11 expression leads to resistance to DNA-damaging agents, including platinum salts (e.g., cisplatin, carboplatin, oxaliplatin) [14,16]. However, we might speculate that since this subgroup of survivors achieved a considerable response to chemotherapy, leading to a reduced tumor burden, the secretion of SLFN11 consequently decreased.

Inflammation is acknowledged to have a role in both the commencement and advancement [18] of tumor formation. Several studies have investigated the link between elevated CRP and poor survival in different cancer types. However, only a limited number of trials, with small sample sizes and different CRP level cutoffs, have explored this link. Based on these studies, it was not possible to draw any solid conclusions [19,20]. The recent Danish study with the highest sample size revealed a link between higher CRP levels and poor survival [21]. Neutrophils and lymphocytes are suggested to have essential roles in tumor-related inflammation [18]. The imbalance between neutrophils and lymphocytes is believed to be the consequence of tumor hypoxia or necrosis and is associated with anti-apoptotic effects [22]. High NLR, based on different cutoff levels, is consistently reported to be linked with poor prognosis in different treatments in several cancers.

Our study’s findings revealed a noteworthy decrease in CRP concentration and leukocyte and neutrophil counts, alongside a simultaneous increase in lymphocyte count from the baseline to the third study point. Consequently, the NLR exhibited a significant decrease from baseline to the second study point. It was evident that patients who experienced earlier mortality during this study initially exhibited the highest levels of CRP, the highest percentage of neutrophils, but the lowest percentage of lymphocytes. Consequently, their NLR was significantly higher compared to the survivors. This group of patients also displayed higher concentrations of SLFN11 compared to the survivors. When comparing survivors with patients who died within two months of this study (at the second study point), we observed significantly higher neutrophil counts in the latter group, along with a higher NLR, while other parameters did not show significant differences.

There are limited studies investigating PD-L1 expression and its prognostic significance in SCLC, and the results are inconsistent [23]. The aggressive nature of SCLC may partly be attributed to its ability to evade the immune system through PD-L1-mediated mechanisms. High levels of PD-L1 expression have been correlated with poorer prognosis, suggesting that immunotherapy targeting PD-L1 may hold promise as a treatment approach for high-grade SCLC [24]. Soluble PD-L1 (sPD-L1) has been linked to poorer survival outcomes, although the precise mechanism is not fully understood. While it is understood that sPD-L1 is a cleavage product of mPD-L1, its biological function in binding PD-1 persists, potentially suppressing and attenuating immune system activity.

Our study results showed that sPD-L1 significantly increased from baseline to the point after two cycles of chemotherapy. We found in the literature that sPD-L1 concentration in SCLC patients was significantly higher in the stable disease group compared to partial responders [25]. This increase in sPD-L1 could be explained by chemotherapy-induced cell death, which can cause the release of tumor-associated antigens and damage-associated molecular patterns. The released molecules can activate the immune system and trigger an inflammatory response. In response to this inflammation, immune cells, such as tumor-infiltrating lymphocytes (TILs), secrete cytokines such as interferon-gamma (IFN-gamma). IFN-gamma is known to stimulate the expression of PD-L1 in tumor cells through the activation of the Janus kinase (JAK) and signal transducer and activator of transcription (STAT) signaling pathway [26]. Another possible mechanism for the increase in PD-L1 expression during chemotherapy is the phenomenon known as adaptive resistance. Tumor cells can undergo adaptive changes in response to treatment, including the upregulation of PD-L1 expression as a survival strategy to evade immune destruction [27]. Also, chemotherapy can trigger an inflammatory response within the tumor microenvironment, which can influence the expression and release of immunomodulatory molecules such as sPD-L1 [28].

In our analysis, to obtain better mortality prediction, we modeled individual parameters and subsequently implemented Kaplan–Meier analysis for combined parameters, i.e., the CRP and NLR combination (inflammatory model), and the CRP, NLR, and SLFN11 combination (inflammatory biomarkers plus SLFN11 model). SCLC patients with higher inflammatory parameters and SLFN11 cutoff levels above 607 ng/mL exhibited significantly lower overall survival. On average, patients in this group had a survival of 3.1 months, whereas those with values below the 75th percentile had an average survival of 4.9 months during the six-month study period of this study.

It is important to recognize the limitations of our study. Specifically, the small sample size may limit the generalizability of our results, thereby necessitating future studies with a larger number of participants to validate our findings. Furthermore, the patients included in this study are exclusively of Caucasian ethnicity, whereas the majority of studies discussed in the literature have focused on patients of Asian ethnicity. This potential population bias may impact the generalizability of the results. Additionally, it is important to highlight the wide range of cutoff values for SLFN11 and sPD-L1 reported in different studies. To ensure accurate interpretation and comparability, it is essential to conduct larger population studies that are prospectively designed for validation purposes. However, the strength of our current study lies in the exploration of novel biomarkers in patients with SCLC that have not been collectively investigated before. These results have the potential to enhance our understanding of the pathophysiological mechanisms underlying SCLC significantly.

## 5. Conclusions

Our findings not only offer valuable insights into the significance of well-known inflammatory markers and two specific protein (PD-L1 and SLFN11) levels after two and four cycles of chemotherapy but also propose a new potential predictive biomarker panel comprising the combination of CRP, NLR, and SLFN11 (inflammatory biomarkers plus the SLFN11 model). Our study revealed that this combination of inflammatory markers could serve as a predictive, non-invasive biomarker for DNA-damaging therapeutic agents in SCLC. Based on the findings, these markers and the proposed panel should be further explored and validated as potential predictive biomarkers for SCLC patient outcomes in significantly larger randomized clinical trials.

## Figures and Tables

**Figure 1 biomedicines-12-00976-f001:**
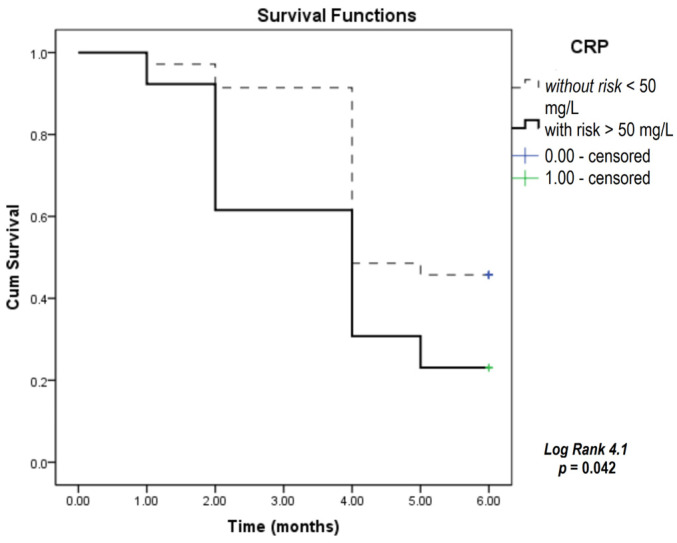
Kaplan–Meier survival analysis conducted in SCLC patients over a six-month study period, focusing on CRP levels. CRP—C-reactive protein.

**Figure 2 biomedicines-12-00976-f002:**
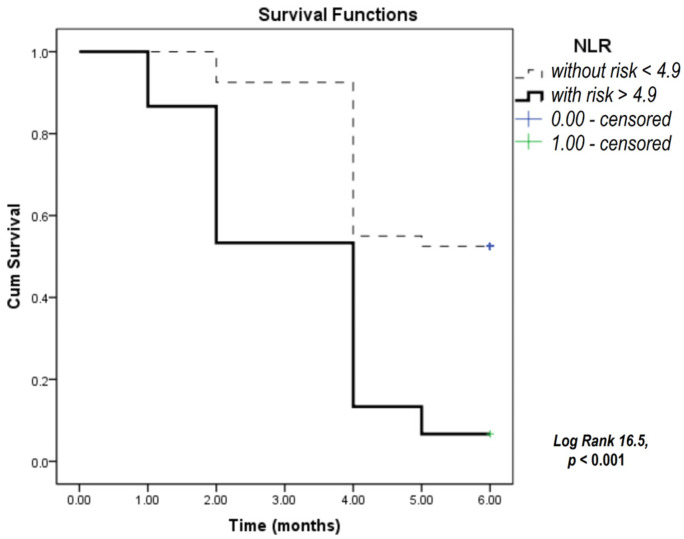
Kaplan–Meier survival analysis conducted in SCLC patients over a six-month study period focusing on the NLR. NLR—neutrophil-to-lymphocyte ratio.

**Figure 3 biomedicines-12-00976-f003:**
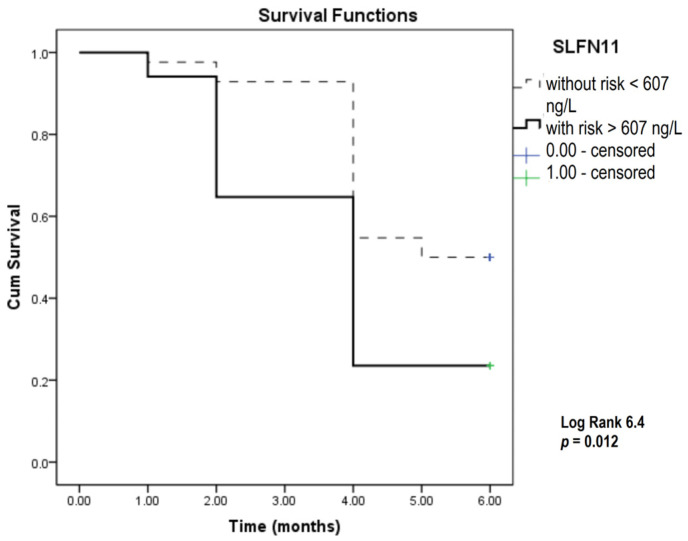
Kaplan–Meier survival analysis in SCLC patients over a six-month study period focusing on SLFN11. SLFN11—Schlafen 11.

**Figure 4 biomedicines-12-00976-f004:**
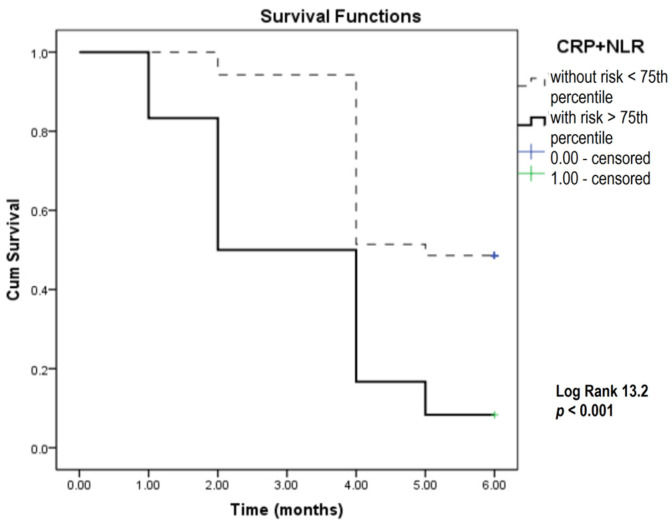
Kaplan–Meier survival analysis in SCLC patients over a six-month study period focusing on CRP and NLR. CRP—C-reactive protein; NLR—neutrophil-to-lymphocyte ratio.

**Figure 5 biomedicines-12-00976-f005:**
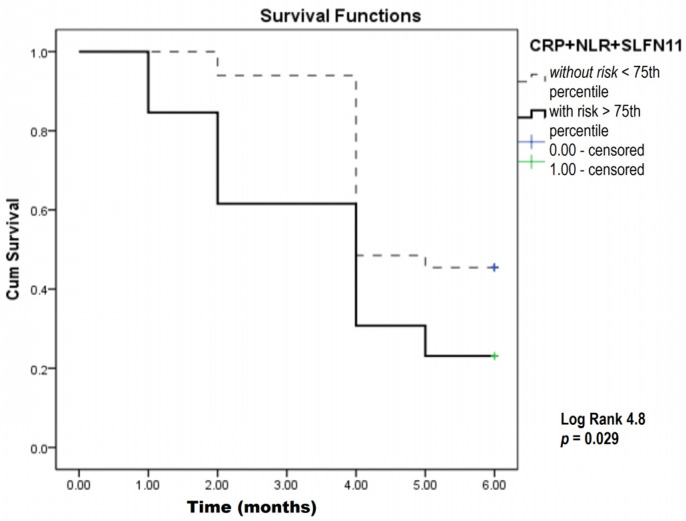
Kaplan–Meier survival analysis in SCLC patients over a six-month study period focusing on CRP, NLR, and SLFN11. CRP—C-reactive protein; NLR—neutrophil-to-lymphocyte ratio; SLFN11—Schlafen 11.

**Table 1 biomedicines-12-00976-t001:** Demographic and clinical data of the study subjects.

Characteristic	Mean ± SD/*n* (%)
Age (in years)	65.5 ± 7.65
Gender: Male/female, *n* (%)	37 (61.6)/23 (38.3)
Smoking status: Never/current/ex-smoker, *n* (%)	1 (1.6)/43 (71.6)/16 (26.6)
TNM staging system:	
Primary tumor size: T 0/1/2/3/4	4 (6.7)/3 (5.0)/10 (16.7)/12 (20.0)/31 (51.7)
Regional lymph nodes: N 0/1/2/3	9 (15.0)/6 (10.0)/31 (51.7)/14 (23.3)
Distant metastases: M 0/1a/1b/1c	32 (53.3)/8 (13.3)/6 (10)/14 (23.3)
Stage III	32 (53.3)
Stage IV	28 (46.6)
Disease stage, *n* (%): LS/ES SCLC	32 (53.3)/28 (46.6)
Distant metastasis, *n* (%)	
Bone	4 (6.6)
Liver	9 (15)
Brain	2 (3.3)
Lung	4 (6.6)
Adrenal gland	9 (15)
Comorbidities, *n* (%)	
COPD	12 (20)
HTA	25 (41.6)
CVDs	16 (26.6)
Arrhythmia	3 (5)
DM	9 (15)
Other malignancies	4 (6.6)
Survival, *n* (%)	
Initial number of patients	60 (100)
after two cycles of chemotherapy	50 (83.3)
after four cycles of chemotherapy	25 (41.7)

European Cooperative Oncology Group Performance Status (ECOG PS); The Veterans’ Administration Lung Study Group (VALSG) Lung Study Group’s 2-stage classification scheme: limited-stage (LS) and extensive-stage (ES) small cell lung cancer (SCLC); chemotherapy (CT); chronic obstructive pulmonary disease (COPD); hypertension (HTA); cardiovascular diseases (CVDs); diabetes mellitus (DM).

**Table 2 biomedicines-12-00976-t002:** Inflammatory parameters, SLFN11, and sPD-L1 in the subgroups according to survival status.

Parameter	Survivors			Deceased		
	Baseline	After 2 Chemotherapy Cycles	After 4 Chemotherapy Cycles	Baseline (Deceased within the First 2 Months of the Study)	Baseline (Deceased within 3rd–4th Months of the Study)	After 2 Chemotherapy Cycles
CRP (mg/L)	9.4 (7.7–15.2)	14.2 (2.9–31.0)	3.0 *^,#^ (2.6–6.8)	95.3 ^aa^ (34.3–160.6)	5.0 ^††^ (1.8–13.6)	3.6 (3.2–16.5)
Leukocyte number (×10^9^/L)	7.7 (6.7–9.4)	7.1 * (5.8–8.3)	6.0 (5.0–8.30)	10.2 (7.9–12.7)	9.2 (7.0–10.0)	7.6 (5.1–10.0)
Neutrophil (%)	68.0 (63.3–69.3)	57.7 ** (53.6–61.8)	65.7 ^#^ (58.2–70.3)	80.8 ^aaa^ (76.7–86.3)	70.1 ^††,b^ (67.7–75.7)	67.7 (50.7–77.1)
Lymphocyte (%)	21.8 (19.4–24.6)	27.6 ** (25.0–32.4)	27.1 * (18.8–31.4)	11.2 ^aa^ (8.2–16.9)	18.9 ^††^ (15.5–24.7)	19.4 (13.0–35.2)
NLR	3.0 (2.5–3.5)	1.9 ** (1.7–2.4)	2.4 ^#^ (1.8–3.8)	7.2 ^aaa^ (4.5–10.5)	3.7 ^††,b^ (2.8–4.9)	3.5 (1.5–5.9)
SLFN11 (pg/mL)	526 (490–544)	1374 *** (1205–1399)	577 ***^,###^ (526–623)	689 ^a^ (575–849)	543 ^†^ (475–596)	1432 ^‡‡^ (1186–1629)
sPD-L1 (pg/mL)	406 (246–727)	567 * (368–940)	525 (261–895)	315 (262–441)	458 (149–677)	456 (283–1108)

*, **, *** *p* < 0.05, 0.01, 0.001 vs. baseline (survivors), respectively; #, ### *p* < 0.05, 0.01 vs. after 2 CT cycles (survivors), respectively. ^†^, ^††^ *p* < 0.05, 0.001, respectively, vs. baseline (deceased in the first 2 months of this study), respectively; ^‡‡^ *p* < 0.01 vs. baseline (deceased in the 3rd–4th months of the study); ^a^, ^aa^, ^aaa^ *p* < 0.05, 0.01, 0.001 vs. baseline (survivors); ^b^ *p* < 0.05 vs. survivors (after 2 chemotherapy cycles). CRP—C-reactive protein; NLR—neutrophil-to-lymphocyte ratio; SLFN11—Schlafen 11; sPD-L1—soluble programmed death ligand 1.

**Table 3 biomedicines-12-00976-t003:** Kaplan–Meier survival analysis data: log-rank coefficients with mean survival times in tested groups.

Log-Rank (Mantel–Cox)			Survival Time (Months)
Survival Predictor	Chi-Square	*p*	
NLR (>4.9 risk)	16.51	<0.001	4.9 ± 0.20 vs. 3.1 ± 0.38
SLFN11 (>607 ng/mL risk)	6.38	0.012	4.9 ± 0.20 vs. 3.7 ± 0.39
CRP (>50 mg/L risk)	4.12	0.042	4.7 ± 0.23 vs. 3.7 ± 0.47
Models			
Inflammatory parameter model * (>75th percentile of predicted probability: risk)	13.17	<0.001	4.9 ± 0.21 vs. 3.1 ± 0.45
Inflammatory + SLFN11 (>75th percentile of predicted probability: risk)	4.75	0.029	4.8 ± 0.21 vs. 3.6 ± 0.49

* Inflammatory parameter model—CRP and NLR. CRP—C-reactive protein; NLR—neutrophil-to-lymphocyte ratio; SLFN11—Schlafen 11.

**Table 4 biomedicines-12-00976-t004:** Risk NLR values and relation with other study parameters in SCLC study patients.

Parameter	NLR < 4.9	NLR > 4.9	*p* *
CRP (mg/L)	8.20 (3.60–22.9)	77.1 (45.9–125.7)	<0.001
WBCs (×10^9^/L)	7.80 (6.65–9.15)	11.10 (8.65–12.20)	0.004
SLFN11 (pg/mL)	524 (465–557)	623 (524–689)	0.026

* *p* Mann–Whitney U test. CRP—C-reactive protein; NLR—neutrophil-to-lymphocyte ratio; WBCs—white blood cells; SLFN11—Schlafen 11.

## Data Availability

Data are available upon request from the corresponding author.

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
