# Peer review of "Correlation of Systemic Inflammation Parameters and Serum SLFN11 in Small Cell Lung Cancer—A Prospective Pilot Study"

_biomedicines, 2024, doi:10.3390/biomedicines12050976_

Round 1
Reviewer 1 Report
Comments and Suggestions for Authors
The authors reported their experience on the correlation between the neutrophil-to-lymphocyte ratio (NLR), c-reactive protein (CRP), soluble programmed cell death ligand 1 (sPD-L1) and Schlafen 11 (SLFN11) with the response to first-line chemotherapy in the cohort of small cell lung cancer patients (SCLC), and to determine their potential as predictive serum biomarkers.
The paper is somehow interesting.
The manuscript is well written.
Some concerns should be addressed by the authors.
- Taking into consideration that patients with stage III disease should receive chemotherapy and concurrent radiotherapy or sequential radiotherapy, I was wondering whether the authors considered this statement or not. No data about RT have been reported. Could the authors comment on that?
- The authors reported that 4 patients were found to have T0 tumor. I am really curious to know some details about those patients. it's a small but intriguing group of patients. Could the authors comment on that?
- Did the authors find any differences between males and females? Did teh authors look at that?
- Looking at the tables and figures reported by the authors, it seems that none of the patients survive more than 6 months. Is that correct? Even considering the well-known poor prognosis related to SCLC, it seems strange that there is not one patient that survive more than 6 months. Could the authors comment on that?
- Is these studies time-consuming? And what about costs? Could the authors comment on that?
-
Author Response
Thank you for dedicating your time to reviewing this manuscript. Below, you will find the detailed responses and the corresponding revisions/corrections highlighted in yellow in the resubmitted manuscript.
- Taking into consideration that patients with stage III disease should receive chemotherapy and concurrent radiotherapy or sequential radiotherapy, I was wondering whether the authors considered this statement or not. No data about RT have been reported. Could the authors comment on that?
We wish to thank the Reviewer for this comment. A clearer explanation about the stage III disease treatment has been provided. All these changes are marked with yellow in the text, in the section 2.2. Patients Selection Criteria (line 143-146). Due to technical limitations at the University Hospital of Pulmonology, Clinical Center of Serbia, concurrent radiotherapy was not available. As a result, all patients with stage III disease underwent sequential chemoradiotherapy, with radiotherapy being administered after the completion of the 4th cycle.
- The authors reported that 4 patients were found to have T0 tumor. I am really curious to know some details about those patients. it's a small but intriguing group of patients. Could the authors comment on that?
We wish to thank the Reviewer for this observation. In response, we have provided further clarification. Four patients classified as having a T0 tumor did not have distinct or separate tumor mass. They presented with a large mediastinal mass solely.
- Did the authors find any differences between males and females? Did teh authors look at that?
We would like to express our appreciation to the Reviewer for this comment. We have provided additional clarification which is highlighted in yellow in the text, specifically in section 3.1. The Dynamics of Inflammatory Parameters Values and sPD-L1 and SLFN11 Concentration Changes in Different Subgroups during the Study (line 208-213). In the survivor group, there was a statistically significant difference in the level of sPD-L1 (p=0.011), with a higher level observed in females. Also, we found statistically significantly higher levels of SLFN11 in females from the deceased group, among those who died within the first 2 months (p=0.030). Additionally, in the deceased group, the levels of SLFN11 remained statistically significantly higher after 2 cycles of chemotherapy (p=0.017). Please find the relevant tables in the attachment.
- Looking at the tables and figures reported by the authors, it seems that none of the patients survive more than 6 months. Is that correct? Even considering the well-known poor prognosis related to SCLC, it seems strange that there is not one patient that survive more than 6 months. Could the authors comment on that?
We cordially thank the Reviewer for this observation. Hence, we have provided additional clarification which has been added to Table 1. Demographic and clinical data of the study subjects. Also, additional clarification highlighted in yellow in the text, specifically in section 3.1. Baseline Characteristics (line 210-203). After two cycles of chemotherapy, 83.3% of patients were alive. However, at the end of the study follow-up period, 41.7% of the patients were alive after four cycles.
|
Survival n (%) Initial number of patients after two cycles of chemotherapy after four cycles of chemotherapy |
60 (100) 50 (83.3) 25 (41.7) |
We apologize for any misunderstanding and for the presence of some typos in the text. We corrected in the text, specifically in 3.2. Survival Analysis Based on Selected Inflammatory Markers (line 248, 255, 258, 261, 269, 272). Our study had a duration of 6 months, and at the end of the study follow-up period, after four cycles of chemotherapy, we had 25 patients (41.7%) who were still alive. These patients attended the control check-up. However, we did not have any further data on these patients as our follow-up period did not extend beyond 6 months from their initial visit (study entry point).
- Is these studies time-consuming? And what about costs? Could the authors comment on that?
We would like to express our gratitude to the Reviewer for this comment and their interest in understanding the logistics behind our study. The enrollment of patients into our study took longer than anticipated. It began in late October 2020, just before the onset of the COVID-19 pandemic, which further complicated the process of recruiting patients. It required meticulous logistics in terms of communication with patients and maintaining comprehensive documentation. The head nurse of the Pulmonology Clinic was responsible for recording and monitoring the collection of blood samples from the 60 enrolled patients: prior to therapy, after two cycles, and after 4th cycle of chemotherapy. Most of the routine biochemical and hematological parameters were obtained from the patients' medical records, without incurring additional costs for analysis. However, there were additional costs associated with specific parameters (SLFN11 and sPD-L1) which required ELISA-type tests. These tests are both expensive and time-consuming, as they involve lengthy procedure performances lasting several hours. Additionally, other parameters (such as redox status parameters, not presented in this current paper) were determined using spectrophotometric tests conducted on an automatic analyzer using in-house made reagents and commercial biochemical control sera. Therefore, the establishment of all these parameters required a certain amount of funding.

Reviewer 2 Report
Comments and Suggestions for Authors
The paper has several strengths, particularly concerning the exploration of novel biomarkers in SCLC patients. The study examines the correlation between various serum biomarkers and the response to first-line chemotherapy in patients SCLC. Key-biomarkers include neutrophil-to-lymphocyte ratio (NLR), c-reactive protein (CRP), soluble programmed cell death ligand 1 (sPD-L1), and Schlafen 11 (SLFN11). The findings presented are valuable and contribute to a better understanding of the pathophysiological mechanisms underlying SCLC.
However, there are some areas that could be improved:
-In the introduction, the authors discuss molecular subtypes associated with SCLC. Did they attempt to apply these subtypes to their own study samples?
-Please provide the normal values for the examined parameters in the M&M section
-Clarify the rationale behind the timing of blood sampling before and after chemotherapy cycles.
-Please specify which organs were affected by distant metastases. Were they in the brain, liver, bones?
-Additionally, it would be beneficial to mention any comorbidities that patients may have alongside the malignancy, potentially influencing certain values
-The authors should discuss potential mechanisms underlying the observed changes in SLFN11 and sPD-L1 levels during chemotherapy cycles
-Moreover, they should provide more detailed information regarding the limitations. They should specify the number of patients for whom measurements were not taken before chemotherapy. This information allows to understand the extent to which the results may be affected by incomplete data collection
-Please also clarify whether you are referring to the predictive or prognostic potential of these biomarkers in SCLC patients undergoing chemotherapy
Comments on the Quality of English Languagemoderate editing. Please check for grammatical and syntactical errors.
Author Response
Thank you for dedicating your time to reviewing this manuscript. Below, you will find the detailed responses and the corresponding revisions/corrections highlighted in yellow in the resubmitted manuscript. English language editing has been completed, and all changes have been indicated in red.
-In the introduction, the authors discuss molecular subtypes associated with SCLC. Did they attempt to apply these subtypes to their own study samples?
We wish to thank the Reviewer for this observation. However, we did not attempt to categorize our study samples based on the molecular subtypes associated with SCLC. The introduction section of the research paper provides a general overview of the molecular subtypes of SCLC, determined by the expression of specific transcriptional regulators like ASCL1, NEUROD1, POU2F3, and YAP1. However, our study focused specifically on investigating the serum levels of SLFN11, systemic inflammation parameters (NLR and CRP), and soluble PD-L1 in SCLC patients who underwent chemotherapy. Our research aimed to analyze the correlation between these parameters and the response to first-line chemotherapy as well as overall survival in SCLC patients. Additionally, it should be noted that in our country, there is no technical capability to perform such an analysis.
-Please provide the normal values for the examined parameters in the M&M section
We would like to express our appreciation to the Reviewer for this comment. We have provided additional clarification which is highlighted in yellow in the text, specifically in section 2.3. Sample Collection and Processing (line 168-173). Considering that most cancer patients typically exhibit elevated inflammation parameters (such as CRP and NLR ratio), we opted not to use normal values as a reference. Instead, based on our statistical analysis, we set the 75th percentile as the high-risk cutoff value for CRP and NLR. It is worth noting that there is no consensus regarding the reference range for SLFN and PDL1. Therefore, we also chose the 75th percentile as the high-risk cutoff value for these parameters.
-Clarify the rationale behind the timing of blood sampling before and after chemotherapy cycles.
We cordially thank the Reviewer for this comment. Hence, we have provided additional clarification which is highlighted in yellow in the text, specifically in section 2.3. Sample Collection and Processing (155-160). The blood samples were collected at multiple time points to monitor the dynamic changes in inflammatory parameters (CRP and NLR), SLFN11 levels, and soluble PD-L1 levels. These measurements taken at baseline, after the second and fourth cycles of chemotherapy, offered insights into the impact of treatment on biomarker levels. They also shed light on potential associations with treatment response and survival, enabling us to evaluate their predictive value over time.
-Please specify which organs were affected by distant metastases. Were they in the brain, liver, bones?
We wish to thank the Reviewer for this comment. We have specified which organs were affected by distant metastasis. All these changes are marked with yellow in the table, in the section 3.1. Baseline Characteristics (Table 1. Demographic and clinical data of the study subject).
|
Distant metastasis n (%) |
Bone |
4 (6.6) |
|
Liver |
9 (15) |
|
|
Brain |
2 (3.3) |
|
|
Lung |
4 (6.6) |
|
|
Adrenal gland |
9 (15) |
-Additionally, it would be beneficial to mention any comorbidities that patients may have alongside the malignancy, potentially influencing certain values
We wish to thank the Reviewer for this comment. We have specified which comorbidities had the investigated group of patients. All these changes are marked with yellow in the table, in the section 3.1. Baseline Characteristics (Table 1. Demographic and clinical data of the study subject).
|
Comorbidities n (%) |
COPD |
12 (20) |
|
HTA |
25 (41.6) |
|
|
CVDs |
16 (26.6) |
|
|
Arrhythmia |
3 (5) |
|
|
DM |
9 (15) |
|
|
Other malignancies |
4 (6.6) |
-The authors should discuss potential mechanisms underlying the observed changes in SLFN11 and sPD-L1 levels during chemotherapy cycles
We cordially thank the Reviewer for this observation. Hence, we have provided additional clarification both for SLFN11 and sPDL1 levels changes, which is highlighted in yellow in the text, specifically in section 4. Discussion, line (364 – 381) and line (416 – 427).
However, previous studies have demonstrated a correlation between SLFN11 hyper-methylation and decreased overall survival (OS) in a cohort of NSCLC patients) [14]. In our study, we observed that patients with a shorter survival time had higher levels of SLFN11. Interestingly, the concentration of SLFN11 significantly increased after two cycles of chemotherapy. However, in the survivor group, it decreased after four cycles of chemotherapy. Nonetheless, even after four cycles of chemotherapy, the level of SLFN11 remained higher than the initial baseline value. Therefore, the increase in SLFN11 levels observed during platinum-based therapy may be due to the activation of the DNA damage response pathways, replication stress in cancer cells, and the cells’ attempt to repair the chemotherapy-induced DNA damage. This upregulation of SLFN11 may contribute to the sensitivity of cancer cells to platinum-based drugs and their subsequent therapeutic effects [18]. Furthermore, we observed a decrease in SLFN11 expression after four cycles of chemotherapy. Several studies confirmed that the loss of SLFN11 expression leads to resistance to DNA-damaging agents, including platinum salts (e.g., cisplatin, carboplatin, oxaliplatin) [14, 17]. However, we might speculate that since this subgroup of survivors achieved a considerable response to chemotherapy, leading to a reduced tumor burden, the secretion of SLFN11 consequently decreased.
This increase in sPD-L1 could be explained as a chemotherapy-induced cell death, which can cause the release of tumor-associated antigens and damage-associated molecular patterns. The released molecules can activate the immune system and trigger an inflammatory response. In response to this inflammation, immune cells, such as tumor-infiltrating lymphocytes (TILs), secrete cytokines such as interferon-gamma (IFN-gamma). IFN-gamma is known to stimulate the expression of PD-L1 in tumor cells through the activation of the Janus kinase (JAK) and signal transducer and activator of transcription (STAT) signaling pathway [28]. Another possible mechanism for the in-crease in PD-L1 expression during chemotherapy is the phenomenon known as adaptive resistance. Tumor cells can undergo adaptive changes in response to treatment, including the upregulation of PD-L1 expression as a survival strategy to evade immune destruction [29].
-Moreover, they should provide more detailed information regarding the limitations. They should specify the number of patients for whom measurements were not taken before chemotherapy. This information allows to understand the extent to which the results may be affected by incomplete data collection
We wish to thank the Reviewer for this comment. We apologize for any potential confusion caused by the sentence, "Also, not all analyses were performed before the start of chemotherapy in all patients." This is related to other to biochemical analysis not analyzed in this particular study, but rather being a part of another project. Therefore, we have made the decision to remove this sentence. However, we provided additional clarification on the limitations of our study, which is highlighted in yellows in the text, specifically in 3.1. Baseline Characteristics (line 196). All patients included in this study were of Caucasian ethnicity. Also, we added additional clarification which is highlighted in yellows in the text, specifically in Discussion, line (440-446). Furthermore, the patients included in this study are exclusively of Caucasian ethnicity, whereas the majority of studies discussed in the literature have focused on patients of Asian ethnicity. This potential population bias may impact the generalizability of the results. Additionally, it is important to highlight the wide range of cutoff values for SLFN11 and sPD-L1 reported in different studies. To ensure accurate interpretation and comparability, it is essential to conduct larger population studies that are prospectively designed for validation purposes.
-Please also clarify whether you are referring to the predictive or prognostic potential of these biomarkers in SCLC patients undergoing chemotherapy
We cordially thank the Reviewer for this observation. We aimed to explore the potential of these parameters (SLFN11, NLR, CRP, sPD-L1) as predictive serum biomarkers for SCLC patients. Hence, we have provided additional clarification which is highlighted in yellow in the text, specifically in section 5. Conclusion, line 454 and 457.
Comments on the Quality of English Language moderate editing. Please check for grammatical and syntactical errors.

Reviewer 3 Report
Comments and Suggestions for Authors
This manuscript examined the correlation between systemic inflammatory parameters and serum SLFN11 in small cell lung cancer (SCLC). The levels of inflammatory markers and two specific proteins (PD-L1 and SLFN11) were found to be significant after 2 and 4 cycles of chemotherapy. But I have some questions for the authors.
1. what was the goal of this manuscript and it is recommended that it be clearly stated in the last part of the INTRODUCTION. parameters such as SLFN11, NLR, CRP and PD-L1 were used to explore their relationship with response to chemotherapy and survival in SCLC patients? Will they be able to find correlations between these parameters and use them as predictive serum biomarkers for SCLC patients?
2. what is the role of SLFN11 in DNA damage response? Needs to be expressed in detail in the discussion.
3.The manuscript has 32% duplication, but I don't think there is plagiarism. But parts of it do need to be rewritten.
Author Response
Thank you for dedicating your time to reviewing this manuscript. Below, you will find the detailed responses and the corresponding revisions/corrections highlighted in yellow in the resubmitted manuscript. The reduction of the duplication rate and English language editing has been completed. All changes have been indicated in red.
- what was the goal of this manuscript and it is recommended that it be clearly stated in the last part of the INTRODUCTION. parameters such as SLFN11, NLR, CRP and PD-L1 were used to explore their relationship with response to chemotherapy and survival in SCLC patients? Will they be able to find correlations between these parameters and use them as predictive serum biomarkers for SCLC patients?
We cordially thank the Reviewer for this observation. A clearer explanation about the goal of the study has been provided. All these changes are marked with yellow in the text, in the Introduction, line (126-128). The goal of this manuscript was to investigate the relationship between SLFN11, NLR, CRP, sPD-L1, response to chemotherapy, and survival outcomes, and potentially use them as predictive serum biomarkers. We collected blood samples at different time points before and during chemotherapy to assess the changes in these biomarker levels and explore their potential correlation with treatment response and overall survival. However, it is crucial to note that the specific conclusions and findings regarding the predictive potential of these biomarkers in SCLC patients are provided in the results and discussion sections of the complete manuscript.
- what is the role of SLFN11 in DNA damage response? Needs to be expressed in detail in the discussion.
We wish to thank the Reviewer for this comment. A clearer explanation about the role of SLFN11 in DNA damage response has been provided. All the changes are marked with yellow in the text, in Discussion, line (348-362). SLFN11 expression has been linked to the sensitivity of cells to various DNA-damaging drugs, including topotecan, cisplatin, and irinotecan. SLFN11 is in-volved in the regulation of biological functions such as cell proliferation, immune responses, and viral replication [14]. In the context of cancer, SLFN11 helps sensitize cancer cells to DNA-damaging agents like topoisomerase I and II inhibitors (such as irinotecan and etoposide, respectively), DNA synthesis inhibitors (e.g., gemcitabine), and DNA cross-linkers and alkylating agents (e.g., cisplatin) [14, 16]. The specific function of the DNA/RNA helicase-like motif in the C-terminal region of SLFN11 is still unknown, but similar motifs have been associated with crucial roles in the cellular response to DNA damage [14]. Previous studies have suggested that SLFN11 interacts with key proteins involved in the DNA damage response system, such as replication proteins RPA1, RPA2, and RPA3, as well as BRCA1-associated ring domain protein (BARD1). Hence, it can be speculated that epigenetic silencing of SLFN11 disrupts the proper interaction between DHX9 and BRCA1, leading to functional changes in the DNA damage response system. This alteration may ultimately affect the sensitivity of cancer cells to platinum-based chemotherapy.
3.The manuscript has 32% duplication, but I don't think there is plagiarism. But parts of it do need to be rewritten.

Round 2
Reviewer 2 Report
Comments and Suggestions for Authors
The authors covered sufficiently all the issues I mentioned.
Comments on the Quality of English LanguageMinor editing